# Probiotic Properties and Neuroprotective Effects of *Lactobacillus buchneri* KU200793 Isolated from Korean Fermented Foods

**DOI:** 10.3390/ijms21041227

**Published:** 2020-02-12

**Authors:** Min-Jeong Cheon, Sung-Min Lim, Na-Kyoung Lee, Hyun-Dong Paik

**Affiliations:** Department of Food Science and Biotechnology of Animal Resources, Konkuk University, Seoul 05029, Korea; alswjd7373@naver.com (M.-J.C.); bluenoah37@nate.com (S.-M.L.); nakyoung_lee@nate.com (N.-K.L.)

**Keywords:** probiotics, kimchi, *Lactobacillus buchneri*, gut-brain-axis, neuroprotective effect

## Abstract

The purpose of this study was to evaluate the probiotic characteristics and neuroprotective effects of bacteria isolated from Korean fermented foods. Three bacterial strains (*Lactobacillus fermentum* KU200060, *Lactobacillus delbrueckii* KU200171, and *Lactobacillus buchneri* KU200793) showed potential probiotic properties, such as high tolerance against artificial gastric juice and bile salts, sensitivity to antibiotics, nonproduction of carcinogenic enzymes, and high adhesion to intestinal cells. Heat-killed *L. fermentum* KU200060 and *L. buchneri* KU200793 showed higher antioxidant activity than heat-killed *L. delbrueckii* KU200171. The conditioned medium (CM) was used to evaluate the reaction between HT-29 cells and each heat-killed strain. All CMs protected SH-SY5Y cells from 1-methyl-4-phenylpyridinium (MPP^+^)-induced toxicity. The expression of brain-derived neurotropic factor (BDNF) mRNA in HT-29 cells treated with CM containing heat-killed *L. buchneri* KU200793 was the highest. The CM significantly reduced the Bax/Bcl*-2* ratio and increased BDNF mRNA expression in SH-SY5Y cells treated with MPP^+^. These results indicate that *L. buchneri* KU200793 can be used as a prophylactic functional food, having probiotic potential and neuroprotective effects.

## 1. Introduction

Probiotics are defined as living microorganisms administered in appropriate amounts that confer a beneficial effect on the host [1]. To be used as probiotics, microorganisms should have the capacity to withstand physical and chemical conditions in the human body and to colonize and adhere to the intestinal epithelial cells [2]. Some probiotics, known as brain probiotics, produce gamma-aminobutyric acid and serotonin [3,4]. Some heat-killed *lactobacilli* alter microbiota composition [5] and affect the host’s neurological and psychiatric functions [6,7].

The conceptual framework of the gut-brain-axis (GBA) has existed for decades [8]. Bidirectional communication between gut microbiota and the components of the GBA influences normal homeostasis and may contribute to neurodegenerative diseases [9]. A recent study showed that gut-brain communication aids in the modulation of immune activity. Chronic proinflammatory immune activity is increasingly recognized as the central element of neurodegenerative disorders, such as Parkinson’s disease (PD) inflammation in the intestine appears particularly related to their pathogenesis [10].

Kimchi is a Korean traditional fermented food, containing biologically active components including polyphenols, vitamins, flavonoids, and lactic acid bacteria (LAB) [11]. Various kinds of microorganisms, including *Leuconostoc* sp., *Lactobacillus* sp., and *Lactococcus* sp., are mainly found in kimchi. Some kimchi bacteria have been verified for antioxidant and immunostimulatory activities [12], antiallergic effects [13,14], and anticancer activity [15].

Some *Lactobacillus buchneri* have been reported for probiotic use [16,17,18]. *L. buchneri* P12 isolated from pickled juice showed cholesterol reduction and antimicrobial activity [17]. *L. buchneri* isolated from kimchi showed the production of high γ-aminobutyric acid (GABA), which as a neurotransmitter, is involved in brain development [16]. In addition, there was production of GABA and ornithine during cheese fermentation [18].

The purpose of this study was to identify *L. fermentum* KU200060, *L. delbrueckii* KU200171, and *L. buchneri* KU200793 isolated from kimchi and to evaluate its probiotic potential, including its tolerance to gastric acid and bile salt conditions, enzyme production, adhesion ability to intestine, and antibiotic susceptibility characteristics. Additionally, this study investigated the neuroprotective effect of heat-killed *L. fermentum* KU200060, *L. delbrueckii* KU200171, and *L. buchneri* KU200793 against MPP^+^- induced cytotoxicity in SH-SY5Y cells.

## 2. Results and Discussion

### 2.1. Tolerance to Artificial Gastric Conditions

The ingested *lactobacilli* can pass through the stomach, which has a low pH environment (pH 2.5–3.5), and then pass through the intestinal tract containing approximately 0.3% (*w/v*) bile acid [19,20]. The tolerance levels of the *Lactobacillus* strains to artificial gastric juice and bile salts are shown in Table 1. The survival rates of *L. fermentum* KU200060 (100.47% ± 0.07%) and *L. buchneri* KU200793 (102.12% ± 0.35%) in gastric acid (0.3% pepsin, pH 2.5) were higher than those of *L. rhamnosus* GG (99.68% ± 0.26%) and *L. delbrueckii* KU200171 (62.68% ± 0.12%). *L. plantarum* 9 showed a survival rate lower than 70% at pH 2.5 and a survival rate higher than 50% at pH 2 [21]. These findings indicate that these isolates have acceptable survival rates in the gastrointestinal environment.

The survival rate of *L. buchneri* KU200793 (91.47% ± 0.46%) slightly decreased, whereas that of *L. rhamnosus* GG (100.41% ± 0.15%), *L. fermentum* 200060 (102.24% ± 0.70%), and *L. delbrueckii* KU200171 (102.43% ± 0.50%) increased under bile salt conditions (0.3% oxgall). Therefore, all the isolated strains pass through the gastrointestinal tract and persist in the intestinal tract. Previous studies suggested that, under acidic conditions, some LAB strains translocated the protons from the cytoplasm to the environment by an ATPase using ATP. So, they modify their cytoplasmic pH. Moreover, some LAB strains have bile salt hydrolase; they could grow with resistance to the bile salt condition [22,23]. According to research on similar strains, the probiotic strains *L. buchneri* 13-2-2 and *L. buchneri* 148-7-1 isolated from human fecal samples could live at pH 3.0 and 0.3% oxgall [24]. *L. buchneri* P2 could live at pH 3.0 for 3 h and in 0.3% oxgall for 14 h [17].

### 2.2. Enzyme Production of LAB Strains

To elicit probiotic properties, no harmful enzymes should be produced by the bacteria. β-glucuronidase is a typical harmful enzyme, which is associated with the induction of toxins, mutagens, and carcinogens [25]. None of the tested strains produced β-glucuronidase, measured using an API ZYM kit (Table 2). However, each strain of *L. rhamnosus* GG, *L. fermentum* KU200060, *L. delbrueckii* KU200171, and *L. buchneri* KU200793 showed the highest production of β-glucosidase, β-galactosidase, leucin arylamidase, and β-galactosidase, respectively. β-Glucosidase, β-galactosidase, and leucin arylamidase are known as beneficial enzymes for hydrolysis of the glycosidic bonds, lactase, and protease, respectively [12].

### 2.3. Antibiotic Susceptibility of LAB Strains

Recent studies demonstrated that commensal bacteria including *Lactobacillus* species can deliver antibiotic resistance genes to another bacteria [26]. The main risk associated with these bacteria is that they can transfer resistance genes to pathogenic bacteria. Consequently, antibiotic susceptibility of probiotics should be tested to determine their safety. All tested LAB strains were resistant to gentamycin, kanamycin, streptomycin, and ciprofloxacin; however, they were susceptible to ampicillin, tetracycline, chloramphenicol, and doxycycline (Table 3). These results are acceptable based on the Clinical and Laboratory Standards Institute guideline [2,27].

### 2.4. Adhesion Ability of LAB Strains to HT-29 Intestinal Epithelial Cells

Adhesion ability of probiotic bacteria to epithelial cells is important for colonization and persistence in the intestinal tract. Moreover, probiotic bacteria that are superior in adhesion ability can competitively bind to the site of adhesion better, preventing the attachment of pathogenic bacteria [27].

To measure the adhesion level of LAB to human intestinal epithelial cells, its adhesion ability to HT-29 cells was analyzed (Table 1). *L. buchneri* KU200793 showed the best adhesion ability (14.10% ± 0.45%) compared to *L. rhamnosus* GG (2.34% ± 0.15%), *L. fermentum* KU200060 (1.18% ± 0.04%), and *L. delbrueckii* KU200171 (1.83% ± 0.01%). *L. paraplantarum* SC61 and *Pediococcus pentosaceus* SC28 have higher adhesion abilities to HT-29 cell (6.26% and 4.03%) than *L. rhamnosus* GG (2.74%) [27]. These results indicate that *L. buchneri* KU200793 had higher adhesion ability to HT-29 cell than other probiotic strains and could attach and colonize the human intestinal epithelial cells.

### 2.5. Antioxidant Effects of Heat-Killed LAB Strains

The results of in vitro antioxidant effects of heat-killed LAB strains are shown in Table 4. *L. buchneri* KU200793 showed DPPH scavenging activity of 23.04% at 10^9^ CFU/mL, which was similar to that of *L. fermentum* KU200060 and *L. rhamnosus* GG. However, *L. delbrueckii* KU200171 showed lower antioxidant effects (17.20%). A stable ABTS radical, which has a blue-green color, is produced by the oxidation of ABTS with potassium persulfate. Radical scavenging activity is measured by the discoloration of ABTS [28]. ABTS radical scavenging activities of *L. rhamnosus* GG (90.92%), *L. fermentum* KU200060 (91.87%), and *L. buchneri* KU200793 (90.05%) were higher than the activity of *L. delbrueckii* KU200171 (68.33%). The inhibition rate of lipid peroxidation in LAB strains was determined by the β-carotene bleaching inhibition assay. *L. buchneri* KU200793 showed the highest inhibition rate of β-carotene and linoleic acid oxidation (38.42%). The inhibition rates were as follows: *L. rhamnosus* GG, 33.63%; *L. fermentum* KU200060, 28.49%; and *L. delbrueckii* KU200171, 16.09%.

As a result, heat-killed *L. fermentum* KU200060 and *L. buchneri* KU200793 showed higher antioxidant activity than heat-killed *L. delbrueckii* KU200171. Heat-killed *L. plantarum* Ln 1 showed 17.60% DPPH radical scavenging activity, 70.18% ABTS radical scavenging activity, and 58.3% inhibition rate of lipid peroxidation at 10^7^ CFU/mL [11]. *L. acidophilus* ATCC 4356 showed 20.8% of DPPH radical scavenging activity in intracellular extract [29]. Antioxidative properties of LAB have been considered as strain-specific depending on the cell wall component, antioxidant enzymes, and exopolysaccharides [20,30]. A recent study referred that oxidative damage could initiate alpha-synuclein, which is a protein involved in Parkinson’s disease [4]. Therefore, these antioxidant effects of heat-killed probiotics may underlie their neuroprotective effects.

### 2.6. Neuroprotective Effects of CM Against MPP^+^ -Induced Cell Death

The cell viability of the conditioned medium (CM) prepared with heat-killed LAB strains *L. rhamnosus* GG, *L. fermentum* KU200060, *L. delbrueckii* KU200171, and *L. buchneri* KU200793 were 107.2% ± 6.0%, 96.0% ± 2.4%, 107.7% ± 2.5%, and 108.6% ± 9.9%, respectively (data not shown). As a result, CM did not show any significant cytotoxicity to SH-SY5Y cells in all the strains.

Treatment of neuroblastoma cells with MPP^+^ increases the levels of reactive oxygen species (ROS), leading to the death of dopaminergic neurons in the *substantia nigra* [31]. The neuroprotective effect of the bacterial strains was determined by assessing the viability of SH-SY5Y cells following MPP^+^ treatment (Figure 1). The viability of SH-SY5Y cells decreased to 61.3% with 1 mM MPP^+^ treatment. However, 4 h pretreatment with CM prepared using heat-killed LAB strains showed cell protective effects. The viability of SH-SY5Y cells following treatment with CM-containing *L. rhamnosus* GG, *L. fermentum* KU200060, *L. delbrueckii* KU200171, and *L. buchneri* KU200793 was 72.0% ± 1.0%, 69.2% ± 0.9%, 66.8% ± 2.2%, and 73.4% ± 0.4%, respectively. These results suggest that CM with LAB strains has potential neuroprotective effects against MPP^+^-treated SH-SY5Y cells. CM using mixed probiotics including *Lactobacillus acidophilus, Bifidobacterium bifidum, Bifidobacterium animalis* subsp. *lactis, Lactobacillus salivarius*, or *Lactobacillus paracasei* has been shown to have neuroprotective effects in SH-SY5Y cells treated with 2 μM MPP^+^ [32].

### 2.7. mRNA Level of the Brain-Derived Neurotropic Factor (BDNF) in HT-29 Cells Treated with Heat-Killed LAB Strains

BDNF is a homodimeric protein with signaling actions mediated via the tyrosine kinase B (trkB) receptor, and it is the most abundant neurotrophic factor in the brain [33]. It has powerful synaptic effects, which promote synaptic transmission, synaptic plasticity, and synaptic growth [34].

To confirm the mRNA expression level of BDNF in HT-29 cells, RT-PCR analysis was performed (Figure 2). The expression levels of BDNF in HT-29 cells treated with heat-killed *L. rhamnosus* GG, *L. fermentum* KU200060, *L. delbrueckii* KU200171, and *L. buchneri* KU200793 were increased by 4.6-, 1.2-, 1.4-, and 5.8-fold compared to the levels in the negative control, respectively. Importantly, treatment with heat-killed *L. buchneri* KU200793 caused the highest increase in BDNF expression. Heat-killed *Ruminococcus albus* has been shown to increase BDNF levels in human intestinal epithelial cells by approximately 1.71-fold [35]. *L. buchneri* KU200793 showed better effects than that observed with *R. albus* treatment in human intestinal epithelial cells.

### 2.8. Bax/Bcl-2 and BDNF Expression Levels of CM in MPP^+^ Stressed SH-SY5Y Cells

BDNF can enhance the survival of dopaminergic neurons and can protect them against the neurotoxic effects of MPP^+^ and mRNA expression of BDNF in the *substantia nigra*, which might affect the death of nigral dopaminergic neurons observed in Parkinson’s disease (PD) [36]. To determine the expression of BDNF in SH-SY5Y cells, RT-PCR was carried out (Figure 3A). Treatment with 1 mM MPP^+^ caused a 0.5-fold reduction in BDNF expression, whereas treatment with the CMs increased BDNF expression levels. *L. rhamnosus* GG and *L. buchneri* KU200793 caused a 1.2- and 1.3-fold increase in BDNF expression compared to that in the negative control, respectively. *L. rhamnosus* GG strain has been reported on neuroprotective potential using cell viability and BDNF mRNA expression in hippocampal neurons [37]. In this study, *L. buchneri* KU200793 has a similar BDNF expression as the *L. rhamnosus* GG strain. Therefore, *L. buchneri* KU200793 can protect MPP^+^-stressed SH-SY5Y cells more effectively.

Bax (Bcl-2-associated X protein), which influences outer membrane permeability, promotes the releases of cytochrome C from the inter membrane of the mitochondria and, finally, induces apoptosis [38]. Bcl-2 (B-cell lymphoma) has antiapoptotic properties and stabilizes membrane permeability, thus preserving mitochondrial integrity, suppressing the release of cytochrome C, and inhibiting apoptosis [39]. Cell survival depends on the balance between pro- and antiapoptotic proteins of the Bcl-2 family. The Bax/Bcl-2 ratio is a better indicator of apoptosis than the absolute concentration of either Bax or Bcl-2 [40]. To investigate the neuroprotective effect of CM in SH-SY5Y cells, Bax and Bcl-2 gene expression levels were evaluated by RT-PCR analysis (Figure 3B). Bax/Bcl-2 mRNA expression ratio increased to 2.3-fold following 1 mM MPP^+^ treatment compared to that in the negative control. However, *L. buchneri* KU200793 significantly decreased the Bax/Bcl-2 ratio to almost normal levels (by 1.0-fold). *L. rhamnosus* GG, *L. fermentum* KU200060, and *L. delbrueckii* KU200171 decreased the ratio by 1.5-, 1.9-, and 2.2-fold, respectively. These results indicate that *L. buchneri* KU200793 was effective in protecting MPP^+^-induced cell apoptosis. The extract of the traditional herb *Gastrodia elata* blume has been shown to effectively reduce the Bax/Bcl-2 mRNA expression ratio in MPP^+^-treated cells. Importantly, pretreatment decreased the Bax/Bcl-2 mRNA expression ratio by 1.5-fold [31]. Compared with previous results, *L. buchneri* KU200793 effectively attenuated apoptosis.

## 3. Materials and Methods

### 3.1. Bacterial Strains Culture Condition

*L. fermentum* KU200060 and *L. buchneri* KU200793 were isolated from watery kimchi and cabbage kimchi, respectively, and *L. delbrueckii* KU200171 was isolated from soy-sauced based fermented crab. All samples (1 g) were serially diluted and plated with using *Lactobacillus* selective medium (BD Biosciences, Franklin Lakes, NJ, USA) and incubated at 37 °C for 24 h. A colony was inoculated and incubated in de Man, Rogosa and Sharpe (MRS; BD Biosciences) broth at 37 °C for 24 h. All strains were identified through 16S rRNA sequencing. *L. rhamnosus* GG was obtained from the Korean Collection for Type Cultures (KCTC; Daejeon, Korea) and was used as a reference probiotic strain. LAB strains were incubated in MRS broth at 37 °C for 24 h. To harvest cells that were intact, bacterial cultures were harvested using centrifuge at 14,240 × *g* at 4 °C for 5 min. The bacterial cells were washed three times and resuspended in phosphate-buffered saline (PBS; Gibco, Grand Island, NY, USA).

### 3.2. Cell Culture Condition

HT-29 (human colon adenocarcinoma; KCLB 30038) and SH-SY5Y (neuroblastoma; KCLB 22266) cells were used for intestine adherence and cytotoxicity experiments, respectively. Each cell lines were cultured in Dulbecco’s modified Eagle’s medium (DMEM; Gibco, Grand Island, NY, USA) at Roswell Park Memorial Institute (RPMI) 1640 (Gibco) containing heat-inactivated 10% fetal bovine serum (FBS; Gibco) and penicillin-streptomycin (Sigma-Aldrich, St. Louis, MO, USA) at 37 °C in a 5% CO_2_ incubator, respectively.

### 3.3. Tolerance to Artificial Gastric Conditions

To measure the tolerance of the strains for the gastric conditions, the experiment was performed according to a previous method [2]. The strains were incubated in MRS broth at 37 °C for overnight. MRS broth with artificial gastric juice (0.3% pepsin (Sigma-Aldrich), pH 2.5) and bile salts (0.3% oxgall (BD Biosciences)) was prepared, and then LAB strains were inoculated at a final concentration of 1 × 10^7^ CFU/mL and incubated at 37 °C for 3 and 24 h, respectively. The viable cells were counted by dilution and plating on MRS agar after each treatment. Survival rate (%) was determined as follows:Survival rate (%) = N_t_/N_i_ ×100(1)

N_i_ and N_t_ represented the viable bacterial cell number before treatment (CFU/mL) and after treatment (CFU/mL), respectively.

### 3.4. Adhesion Ability to HT-29 Cells

The adhesion ability of LAB strains was evaluated using HT-29 cells. HT-29 cells were cultured in a 24-well plate (final concentration: 1 × 10^5^ cells/well) and incubated at 37 °C for overnight. After incubation, LAB strains (1 × 10^8^ CFU/well) were cultured with HT-29 cells for 37 °C for 2 h. Nonadherent bacterial cells were washed three times with PBS. Then, 1% Triton X-100 (Sigma-Aldrich) solution was treated to detach the adherent bacterial cells. The number of adherent bacterial cells to the HT-29 was evaluated by serial dilutions, spreading on MRS agar, and incubating at 37 °C for 24 h. Finally, there was counting of viable cells on MRS plates. Adhesion ability was also determined using Equation (1).

### 3.5. Enzyme Production Evaluated Using API ZYM Kit

Enzyme production was evaluated using API ZYM kit (bioMérieux, Marcy-l’Étoile, France). Bacterial samples (70 μL) were inoculated in each cupule and incubated at 37 °C for 4 h. After incubation, ZYM A and ZYM B reagents were added to each cupule in turn. Enzyme production was evaluated using color intensity scores from 0 (no activity) to 5 (≥40 nM).

### 3.6. Antibiotic Susceptibility

Antibiotic susceptibility of LAB strains was evaluated using the disk diffusion method according to the Clinical and Laboratory Standards Institute (CLSI) guidelines [27]. Susceptibility to the following eight antibiotics was evaluated: 0.2 mg/mL of ampicillin, gentamicin, and streptomycin; 0.6 mg/mL of kanamycin, tetracycline, chloramphenicol, and doxycycline; and 0.1 mg/mL of ciprofloxacin. One hundred-fifty microliters of each strain (1 × 10^6^ CFU/mL) was spread on an MRS agar plate, and paper discs were set on the plate. Then, the antibiotics were loaded on the paper disc and incubated at 37 °C for overnight. After incubation, the inhibition zone (mm) was measured.

### 3.7. Preparation of Heat-Killed LAB

Bacteria cultures (1 × 10^9^ CFU/mL) were harvested by centrifugation at 14,240 × *g* at 4 °C for 5 min, and the supernatant was removed. The bacterial cells were washed three times and were resuspended in phosphate-buffered saline. Then, the washed bacteria were heated at 121 °C for 15 min.

### 3.8. Antioxidant Activity of Heat-Killed LAB Strains

The antioxidant activity of the heat-killed LAB strains was measured by 2,2-diphenyl-1-picrylhydrazyl (DPPH) assay. The assay was evaluated according to a previous method with some modifications [12]. DPPH solution of 400 μM was prepared using ethanol, and 500 μL of heat-killed LAB samples or PBS (control) was mixed with 500 μL of this solution. The mixture reacted in dark conditions at 37 °C for 30 min. The absorbance of the mixture was measured at 517 nm and calculated using Equation (2).
Radical scavenging activity (%) = (1-A_sample_/A_control_) × 100(2)

The radical scavenging activity of 2,2′-azinobis (2-ethylbenzothiazoline-6-sulfonate) (ABTS) was measured as described by a previous method with some modifications [2]. To prepare ABTS solution, 7 mM ABTS and 5 mM potassium persulfate was mixed with 20 mM sodium phosphate buffer (pH 7.4), and they were allowed to react at room temperature in the dark for 16 h. Using 20 mM sodium phosphate buffer, ABTS^+^ solution was diluted until the final absorbance changed to 0.7 ± 0.02 at 734 nm. Thereafter, 100 μL of heat-killed bacterial samples and PBS (control) were mixed with 900 μL of ABTS^+^ solution and reacted at 37 °C for 10 min in the dark condition. The percentage scavenging of ABTS radicals was measured at 734 nm and was calculated using Equation (2).

The β-carotene bleaching inhibition assay was measured as described by a previous method with some modifications [20]. β-carotene solution was prepared using 3 mg of β-carotene, 66 μL of linoleic acid, 300 μL of Tween 80, and 10 mL of chloroform. The solution was evaporated to remove chloroform at 40 °C under vacuum using a rotary evaporator and were diluted with 75 mL of distilled water. Heat-killed bacterial samples (200 μL) and control (PBS, 200 μL) were mixed immediately with 4 mL of the emulsion and were reacted in a water bath at 50 °C for 2 h. Finally, the absorbance of reacted solution was measured at 470 nm for 0 and 2 h. Inhibition of β-carotene and linoleic acid oxidation was calculated as the following formula:

Inhibition of β-carotene and linoleic acid oxidation (%) = (A_S,2 h_ -A_C,2 h_) / (A_C,0 h_ -A_C,2 h_) × 100

A_C_ and A_S_ represent the absorbance of control and sample with each time, respectively.

### 3.9. Preparation of LAB-Conditioned Medium (CM)

HT-29 cells were plated in 6-well plates at a concentration of 5 × 10^5^ cells/mL and were incubated to form a monolayer. Next, heat-killed LAB strains and PBS (for control) treated the cell and incubated 24 h. After 24 h, the cell supernatant was centrifugated at 14,000 × *g* for 5 min. Subsequently, the supernatant was filtered with a 0.45 μm pore size syringe filter and was stored at −80 °C.

### 3.10. Cytotoxicity Measurement

#### 3.10.1. Cytotoxicity Effect of CM on SH-SY5Y Cell

The 3-(4,5-dimethylthiazol-2-yl)-2,5-diphenyltetrazolium bromide (MTT) assay was used to measure the effect of CM on the survival rates of SH-SY5Y cells. SH-SY5Y cells were plated in 96-well plates at a concentration of 1 × 10^5^ cells/well and incubated 24 h. After overnight, CM was applied in the SH-SY5Y cells. After then, MTT dissolved in PBS solution (5 mg/mL) was added to the cells, and the plates were incubated at 37 °C for 1 h. The solution was removed, and 100 μL of DMSO was added to each well. After 15 min, the absorbance was measured at 540 nm using an ELISA reader (Molecular Devises, Sunnyvale, CA, USA), and viability was calculated as in Equation (3).
Cell viability (%) = As/Ac × 100(3)

As and Ac represent the absorbance of sample treatment and control without sample, respectively.

#### 3.10.2. Neuroprotective Effect of CM on MPP^+^-stressed SH-SY5Y Cells.

Cell cytotoxicity was induced using 1-methyl-4-phenylpyridinium (MPP^+^) (Sigma-Aldrich, St. Louis, MO, USA) to determine the protective effect of CM on SH-SY5Y cells. Cells (1 × 10^5^ cells/well) were seeded into 96-well plates. Then, cells were treated with CM for 4 h, and subsequently, MPP^+^ (1 mM) was used in treating the cells for 20 h.

The media were removed, and MTT solution (5 mg/mL) was added and incubated for 1 h. Next, the solution was removed, and 100 μL DMSO was added to each well. Absorbance was measured at 540 nm using an ELISA reader. Then, cell viability (%) was calculated using Equation (3).

### 3.11. Relative Quantification of Gene Expression by RT-PCR

HT-29 and SH-SY5Y cells were plated in 6-well plates at 1 × 10^6^ cells/well and were incubated for 7 days and 48 h, respectively. BDNF expression in HT-29 cells was induced by inoculation of 1 × 10^8^ CFU/well of heat-killed cells and incubated for 24 h. To measure the neuroprotective effect of CM, SH-SY5Y cells were applied for 4 h using CM, and subsequently, 1 mM MPP^+^ was used in treating the cells for 20 h.

Total RNA was using RNeasy^®^ Mini Kit (QIAGEN, Hilden, Germany) and cDNA was synthesized using SensiFAST™ cDNA Synthesis Kit (Bioline, London, UK). The expression levels of cytokines related to cell apoptosis and the neurotrophic factor were determined using SYBR Green PCR Master mix with real-time PCR (PikoReal 96, Scientific Pierce, Waltham, MA, USA). The primers are listed in Table 5. Real-time PCR was performed at 95 °C for 2 min for initial denaturation, and subsequently, 40 cycles were performed as the following conditions: denaturation at 95 °C for 5 s and 60 °C for 15 s for annealing/extension. The results were analyzed using the delta-delta Cq method. The melting curve was used for analyses to measure reaction specificity.

### 3.12. Statistical Analysis

All experiments were repeated in triplicates, and one-way analysis of variance (ANOVA) and Duncan’s multiple range tests were conducted using SPSS software (Version 18; SPSS Inc., Chicago, IL, USA) to measure significant differences (*p* < 0.05). Data are presented as mean ± standard error.

## 4. Conclusions

*L. fermentum* KU200060, *L. delbrueckii* KU200171, and *L. buchneri* KU200793 isolated from Korean fermented foods have probiotic potential. These bacteria showed high gastric acid and bile salt tolerance, safe enzyme activity, acceptable antibiotic susceptibility, and a high HT-29 adhesion ability. Among the three strains, heat-killed *L. buchneri* KU200793 showed relatively good antioxidant activity. Cells treated with CM containing heat-killed *L. buchneri* KU200793 had the highest levels of BDNF and survival rates. *L. buchneri* KU200793 can be used as a functional food ingredient to prevent Parkinson’s disease. However, further in vivo studies are needed to verify that *L. buchneri* KU200793 is effective as a neuroprotective agent.

## Figures and Tables

**Figure 1 ijms-21-01227-f001:**
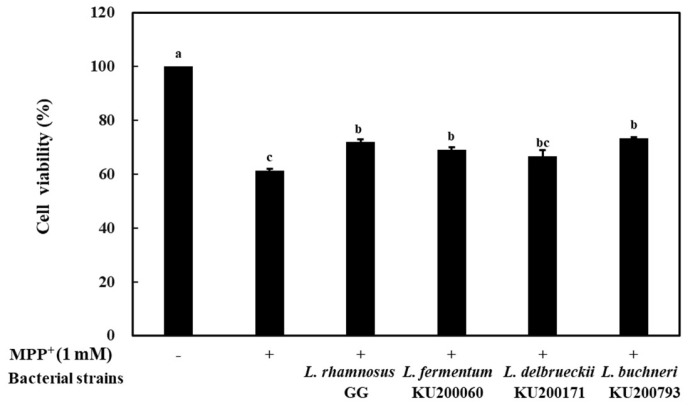
Neuroprotective effect of conditioned medium (CM) on MPP^+^ (1 mM)-stressed SH-SY5Y cells. *L. rhamnosus* GG, CM of heat-killed *L. rhamnosus* GG; *L. fermentum* 200060, CM of heat-killed *L. fermentum* KU200060; *L. delbrueckii* 200171, CM of heat-killed *L. delbrueckii* KU200171; and *L. buchneri* 200793, CM of heat-killed *L. buchneri* KU200793. All values are expressed as mean ± standard error of triplicate experiments. Different superscript letters on each bar represent significant differences (*p* < 0.05).

**Figure 2 ijms-21-01227-f002:**
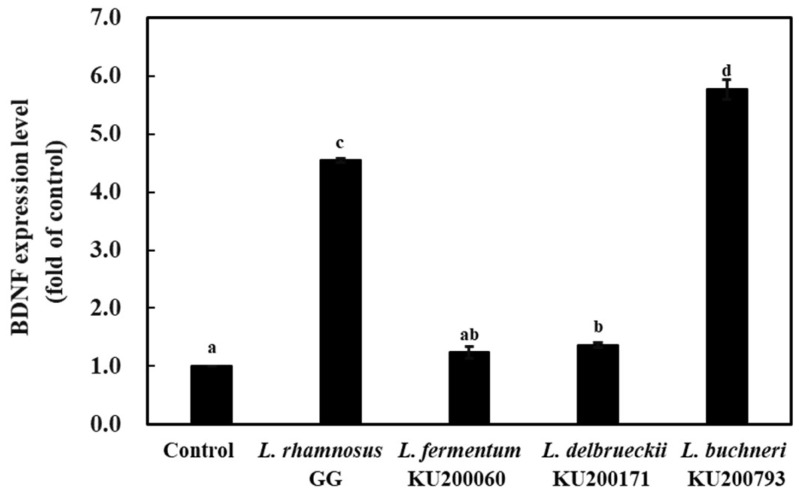
mRNA expression levels of the brain-derived neurotropic factor (BDNF) gene in heat-killed lactic acid bacteria (LAB) treated HT-29 cells using RT-PCR. The fold change was calculated based on normalization with GAPDH gene expression. Control group was treated with PBS. *L. rhamnosus* GG, heat-killed *L. rhamnosus* GG; *L. fermentum* 200060, heat-killed *L. fermentum* KU200060; *L. delbrueckii* 200171, heat-killed *L. delbrueckii* KU200171; and *L. buchneri* 200793, heat-killed *L. buchneri* KU200793. All values are expressed as mean ± standard error of triplicate experiments. Different superscript letters on each bar represent significant differences (*p* < 0.05).

**Figure 3 ijms-21-01227-f003:**
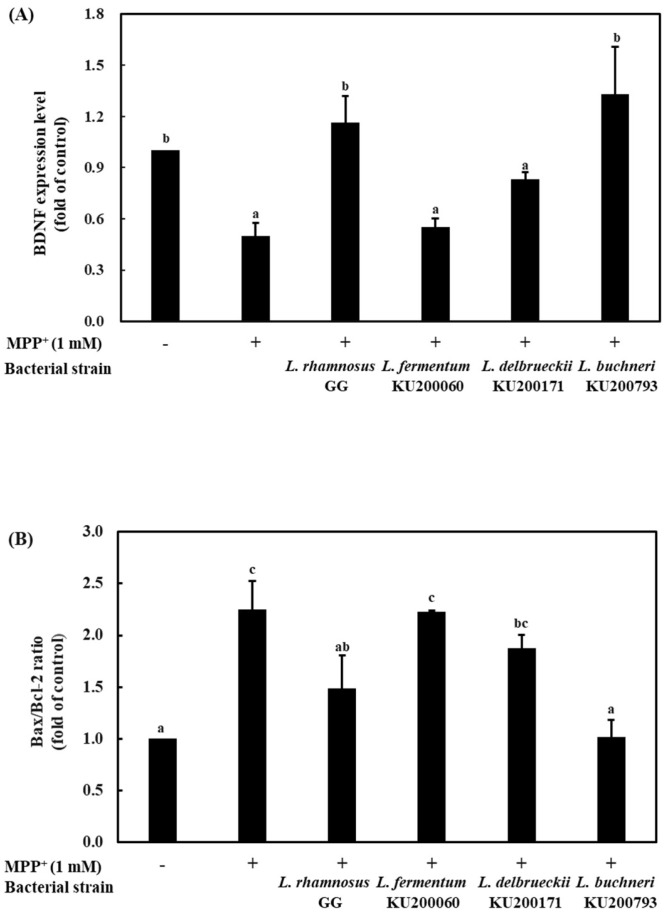
mRNA expression of (**A**) BDNF and (**B**) apoptosis-related genes (Bax/Bcl-2 ratio) in SH-SY5Y cells using RT-PCR. *L. rhamnosus* GG, CM of heat-killed *L. rhamnosus* GG; *L. fermentum* 200060, CM of heat-killed *L. fermentum* KU200060; *L. delbrueckii* 200171, CM of heat-killed *L. delbrueckii* KU200171; and *L. buchneri* 200793, CM of heat-killed *L. buchneri* KU200793. All values are mean ± standard error of triplicate experiments. Different superscript letters on each bar represent significant differences (*p* < 0.05).

**Table 1 ijms-21-01227-t001:** Tolerance of *Lactobacillus* strains to artificial gastric conditions and its adhesion ability to the HT-29 cell.

LAB Strains	Survival Rate (%)		Adhesion Ability to HT-29 Cells (%)
	Gastric Acid Tolerance (0.3% Pepsin, pH 2.5)	Bile Salts Tolerance (0.3% Oxgall)	
*L. rhamnosus* GG	99.68 ± 0.26 ^c (1)^	100.41 ± 0.15 ^b^	2.34 ± 0.15 ^b^
*L. fermentum* KU200060	100.47 ± 0.07 ^b^	102.24 ± 0.70 ^a^	1.18 ± 0.04 ^c^
*L. delbrueckii* KU200171	62.68 ± 0.12 ^d^	102.43 ± 0.50 ^a^	1.83 ± 0.01 ^bc^
*L. buchneri* KU200793	102.12 ± 0.35 ^a^	91.47 ± 0.46 ^c^	14.10 ± 0.45 ^a^

^(1) a–d^ Different superscript letters in the same row indicate significant differences in each characteristic (*p* < 0.05). All values are mean ± standard error of triplicate experiments.

**Table 2 ijms-21-01227-t002:** Enzyme activities of the *Lactobacillus* strains measured using the API ZYM kit.

Enzyme	Enzyme Activity			
	*L. rhamnosus* GG	*L. fermentum* KU200060	*L. delbrueckii* KU200171	*L. buchneri* KU200793
Control	0 ^(1)^	0	0	0
Alkaline phosphate	0	0	0	0
Esterase	2	1	0	0
Esterase lipase	1	1	0	0
Lipase	0	0	0	0
Leucine arylamidase	3	2	4	3
Valine arylamidase	3	0	3	2
Cystine arylamidase	0	0	0	0
Trypsin	0	0	0	0
α-Chymotrypsin	0	0	0	0
Acid phosphatase	1	1	1	0
Naphthol-AS-BI-phosphohydrolase	2	1	2	1
α-Galactosidase	0	4	0	0
β-Galactosidase	0	5	0	4
β-Glucuronidase	0	0	0	0
α-Glucosidase	3	1	0	0
β-Glucosidase	4	0	0	1
*N*-Acetyl-β-glucosaminidase	0	0	0	2
α-Mannosidase	0	0	0	0
α-Fucosidase	0	0	0	0

^(1)^ 0, 0 nM; 1, 5 nM; 2, 10 nM; 3, 20 nM; 4, 30 nM; and 5, ≥40 nM.

**Table 3 ijms-21-01227-t003:** Antibiotic resistances of the *Lactobacillus* strains.

Antioxidant Activity	LAB Strains			
	*L. rhamnosus* GG	*L. fermentum* KU200060	*L. delbrueckii* KU200171	*L. buchneri* KU200793
Ampicillin	S ^(1)^	S	S	S
Gentamicin	R	R	R	R
Kanamycin	R	R	R	R
Streptomycin	R	R	R	R
Tetracycline	S	S	S	S
Ciprofloxacin	R	R	R	R
Chloramphenicol	S	S	S	S
Doxycycline	S	S	S	S

^(1)^ S, susceptible; I, intermediate; and R, resistant.

**Table 4 ijms-21-01227-t004:** Antioxidant activities of the *Lactobacillus* strains.

Antioxidant Activity	LAB Strains			
	*L. rhamnosus* GG	*L. fermentum* KU200060	*L. delbrueckii* KU200171	*L. buchneri* KU200793
DPPH radical scavenging activity (%)	23.76 ± 1.53 ^a (1)^	21.81 ± 1.62 ^a^	17.20 ± 0.48 ^b^	23.04 ± 0.88 ^a^
ABTS radical scavenging activity (%)	90.92 ± 1.91 ^a^	91.87 ± 3.27 ^a^	68.33 ± 1.91 ^b^	90.05 ± 3.27 ^a^
Inhibition rate of β-carotene and linoleic acid oxidation (%)	33.63 ± 2.60 ^ab^	28.49 ± 2.79 ^b^	16.09 ± 2.13 ^c^	38.42 ± 3.56 ^a^

^(1) a–c^ Different superscript letters in the same row indicate significant differences in each characteristic (p < 0.05). All values are the mean ± standard error of triplicate experiments.

**Table 5 ijms-21-01227-t005:** Primer sequences related to the neuroprotective effect used in real-time PCR.

Primer ^(1)^	Sequence (5′-3′)	Reference
BDNF	(sense) CAAACATCCGAGGACAAGGTGG	[35]
	(antisense) CTCATGGACATGTTTGCAGCATCT	
Bax	(sense) GTGGTTGCCCTCTTCTACTTTGC	
	(antisense) GAGGACTCCAGCCACAAAGATG	
Bcl-2	(sense) CGGCTGAAGTCTCCATTAGC	
	(antisense) CCAGGGAAGTTCTGGTGTGT	
GAPDH	(sense) GAGTCAACGGATTTGGTCGT	
	(antisense) GACAAGCTTCCCGTTCTCAG	

^(1)^ BDNF, brain-derived neurotrophic factor; Bax, bcl-2-associated X protein; Bcl-2, B-cell lymphoma; and GAPDH, glyceraldehyde-3-phosphate dehydrogenase.

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
