# Peer review of "Probiotic Properties and Neuroprotective Effects of Lactobacillus buchneri KU200793 Isolated from Korean Fermented Foods"

_ijms, 2020, doi:10.3390/ijms21041227_

Round 1

Reviewer 1 Report

In this manuscript, Cheon et al. examined the probiotic properties and neuroprotective effects of Lactobacillus buchneri and some other LAB. The study was well conducted and the results were presented clearly. I have some minor suggestions to potentially improve the manuscript:

I would like to see the data for antibiotic susceptibility, at least in a table format. Please use the names of the strains instead of the numbers in the tables and figures. The numbers can be indicated in the figure legend. The method describing the isolation of L. fermentum, L. buchneri and L. delbruechii is too brief. More information should be included. I am curious why As,0h was not used in the equation for b-carotene bleaching assay? How much differences were found between As,0h and Ac,0h? Please Add “, respectively” after “However, each strain of L. rhamnosus GG, L. fermentum KU200060, L. delbrueckii KU200171, and L. buchneri KU200793 showed highest production of β-glucosidase, β-galactosidase, leucin arylamidase, and β-galactosidase” in Line 84. Some descriptions in the Materials and Methods are very similar to those in the literature. Please change the wording. Citations are missing for this sentence: “BDNF can enhance the survival of dopaminergic neurons and can protect them against the neurotoxic effects of MPP+” in lines 172-173. In line 47-48, it says “L. buchneri isolated from kimchi showed the production of high γ-aminobutyric acid (GABA) related neurotransmitter, brain development, and nervous system.” I don’t quite understand this sentence. Are there some words missing? Similar issue with “Commensal bacteria may act as reservoirs of antibiotic resistance genes like human pathogens” in Line 91.

Author Response

Dear reviewers:

We thank you for your review and comments to improve the quality of our manuscript. Please find enclosed, a revised copy of the manuscript ID: ijms-708141. We hope the manuscript is now ready for publication.

Thank you,

Prof. Hyun-Dong Paik

Department of Food Science and Biotechnology of Animal Resources

Konkuk University

Seoul, Korea

Reviewer #1:

I would like to see the data for antibiotic susceptibility, at least in a table format.

→ As your comments, we inserted in table 3.

Please use the names of the strains instead of the numbers in the tables and figures. The numbers can be indicated in the figure legend.

→ As your comments, we revised all tables and figures.

The method describing the isolation of L. fermentum, L. buchneri and L. delbruechii is too brief. More information should be included.

→ As your comments, we added more information about isolation of LAB strains in line 223-228.

I am curious why As,0h was not used in the equation for b-carotene bleaching assay? How much differences were found between As,0h and Ac,0h?

→ The formula for the b-carotene bleaching assay is the percentage of the natural oxidation of the control group (0 h to 2 h) versus the time of oxidation when the sample and the control were reacted at the same time (2 h). In addition, the difference between As, 0h and Ac, 0h is not large because it does not react quickly. Therefore, in this study, we did not use As,0h in the equation.

Please Add “, respectively” after “However, each strain of L. rhamnosus GG, L. fermentum KU200060, L. delbrueckii KU200171, and L. buchneri KU200793 showed highest production of β-glucosidase, β-galactosidase, leucin arylamidase, and β-galactosidase” in Line 84.

→ Thank you for your comments, we added “respectively” in line 86.

Some descriptions in the Materials and Methods are very similar to those in the literature. Please change the wording.

→ Thank for your comments, we revised the Materials and Methods.

Citations are missing for this sentence: “BDNF can enhance the survival of dopaminergic neurons and can protect them against the neurotoxic effects of MPP+” in lines 172-173.

→ It citated same paper. So, we revised sentence clearly in line 186.

In line 47-48, it says “L. buchneri isolated from kimchi showed the production of high γ-aminobutyric acid (GABA) related neurotransmitter, brain development, and nervous system.” I don’t quite understand this sentence. Are there some words missing?

→ Thank you for your comments, we corrected the sentence more clearly in line 46-48.

Similar issue with “Commensal bacteria may act as reservoirs of antibiotic resistance genes like human pathogens” in Line 91.

→ We revised the sentence in line 96-97.

Reviewer 2 Report

Interesting study but the presentation of the results and some methods used are not appropriate.

Use italics in strain names etc.

Line 59. “Probiotic bacteria can thrive in the gut as they are resistant to gastric acid and bile salts”. Are you sure that this applies to all probiotic microorganisms?

Lines 68-71. No meaning.

Table 1. Please explain the accuracy of two decimals in the survival of cells.

Table 1. Please explain the increase observed in solutions with no nutrient and under harsh conditions.

Line 223. Please explain in details. The previous study [2], refers to another previous study that also refers to a previous study that used nutrient broth. Does this also apply to the present study or only solutions in water were used?

Lines 226-228. Using agar? If yes which one?

Line 228. Reaction?? Of what? Probably treatment? Incubation?

Line 228. Ni and Nt are in CFU/mL or logCFU/mL?

Lines 221-228. Please explain why separate treatments were used? This process tries to simulate the gastrointestinal conditions and someone would expect the same cells first to incorporate in gastric juice and then (the same cells) to incorporate in bile salts. The passage through gastric juice may lead to cell injuries that will affect the subsequent passage through bile salts. Please explain in details.

Line 254. Please explain how the exact number of died cells were measured. In addition the use of CFU is wrong since we have died cells.

Line 329. Probiotic food? I do not think that a bacteria is food…

Conclusions need revision. Some future studies should be proposed and the importance of the results should be highlighted.

The authors do not discuss their results with those of other similar studies using similar strains.

In my opinion the use of separate incubation of live cells under simulated gastric juice and bile salts is a serious problem of the study.

Author Response

Line 59. “Probiotic bacteria can thrive in the gut as they are resistant to gastric acid and bile salts”. Are you sure that this applies to all probiotic microorganisms?

→ We do not think high survival rate of probiotics. Their survival rated depends on species. Therefore, we confirmed the survival rate.

Lines 68-71. No meaning.

Lactobacillus buchneri strain used in the manuscript is a less studied strain, especially only a fewer strains has probiotic properties. Therefore, the resistant rate of artificial gastric condition measured in the manuscript and the Lactobacillus buchneri in other papers are referenced for comparison. To clarify the wording, we revised the sentence in line 73-75.

Table 1. Please explain the accuracy of two decimals in the survival of cells.

→ The survival rate calculated the ratio of cell no. of initial and after reaction having three decimals. We arranged the accuracy of two decimals. We tested triplicate experiments; we suggested the value of mean ± standard error. In addition, we suggested the significancy of p < 0.05.

Table 1. Please explain the increase observed in solutions with no nutrient and under harsh conditions.

→ Patel et al referred that majority of species from Lactobacillus has bile salt hydrolase, so they grow with resistance to the bile salt media.

Charalampopoulos et al. suggested that under acidic conditions, some LAB strains translocated the protons from the cytoplasm to the environment by an ATPase at the expense of ATP. And then, they modify their cytoplasmic pH. When L. plantarum and L. acidophilus exposed at acidic conditions (pH 3 and 3.5), researchers observed that high ATPase activity. We discussed in line 69-72.  

Line 223. Please explain in details. The previous study [2], refers to another previous study that also refers to a previous study that used nutrient broth. Does this also apply to the present study or only solutions in water were used?

→ In this manuscript, we used MRS broth and revised the sentence in line 241-242.

Lines 226-228. Using agar? If yes which one?

→ We used MRS agar. The sentence revised in line 243-244.

7.Line 228. Reaction? Of what? Probably treatment? Incubation?

→ We revised reaction to treatment. The sentence corrected in line 246-247.

Ni and Nt are in CFU/mL or logCFU/mL?

→ Ni and Nt are CFU/mL. The sentence revised in line 246-247.

Lines 221-228. Please explain why separate treatments were used? This process tries to simulate the gastrointestinal conditions and someone would expect the same cells first to incorporate in gastric juice and then (the same cells) to incorporate in bile salts. The passage through gastric juice may lead to cell injuries that will affect the subsequent passage through bile salts. Please explain in details.

→ Some researcher investigated the subsequent passages. However, many researchers tested as each step of gastric condition having acid conditions (pH 1.5~3.0) and bile conditions. Especially, these strains have high survival rate, these results may ignore the loss of subsequent passage.

Line 254. Please explain how the exact number of died cells were measured. In addition the use of CFU is wrong since we have died cells.

→ There was some error in the wording. When the strains were grown for 24 hours, the concentration reached to 1 × 109 CFU/ml and then, the bacteria cultures were harvested and heat treatment. We revised in line 270.

Line 329. Probiotic food? I do not think that a bacteria is food…

→ We revised in line 349.

Conclusions need revision. Some future studies should be proposed and the importance of the results should be highlighted.

→ As your comments, we suggested their importance on prophylactic effect of Parkinson’s diseases, and future studies.

13.The authors do not discuss their results with those of other similar studies using similar strains.

→ Although L. buchneri strains was reported to have probiotic properties, the number is very few and there are very few cases of neuroprotective effects among the probiotic L. buchneri strains.

LGG strains showed increased cell viability and BDNF m RNA expression in neuron cell (Cheng et al., 2019) Therefore, this result added in line 188-191.

In my opinion the use of separate incubation of live cells under simulated gastric juice and bile salts is a serious problem of the study.

→ This study was suggested probiotic and prophylactic properties of isolated strains. In addition, in vitro test, neuroprotective effect was treated on conditioned medium (CM) using heat-killed cells. In addition, in vitro test, we can use live cells and confirmed prophylactic effect of Parkinson disease. In this point, we can represent the two type studies.

Reviewer 3 Report

The publication presented to me for review is part of the trend of searching for new probiotic strains with significant pro-health activities. The authors isolated several strains of lactic acid bacteria from kimchi and tested them for probiotic properties as well as properties that could potentially result in their neuroprotective effects.

The publication is consistent and properly prepared. The methodology is presented in a simple but sufficient way. A certain dissatisfaction is left only by the discussion of the results, which in my opinion is too short. I would like to ask authors for a broader reference to the results presented by other authors (e.g. when describing antioxidant properties they refer to only one species, which in addition is not among the bacteria described in their research. The importance of the antioxidant LAB effect is described in one sentence, please expand the topic. Please expand the entire discussion.)

I also have some editorial comments: the names of the species are not italics in almost the entire text, please correct it. Same with "substantia nigra".

Line 105: please expand P. pentosaceus (Pediococcus pentosaceus)

Line 283: It should be -80 instead of _80

Line 329: I think it would be better to expand the abbreviation PD (Parkinson’s disease)

Despite the above remarks, I find the publication interesting and worth publishing (if the authors develop the discussion of the results).

Author Response

The publication is consistent and properly prepared. The methodology is presented in a simple but sufficient way. A certain dissatisfaction is left only by the discussion of the results, which in my opinion is too short. I would like to ask authors for a broader reference to the results presented by other authors (e.g. when describing antioxidant properties, they refer to only one species, which in addition is not among the bacteria described in their research. The importance of the antioxidant LAB effect is described in one sentence, please expand the topic. Please expand the entire discussion.)

→ Thank you for your comment, we added the discussions in line 68-74, 128-132, 188-191.

The names of the species are not italics in almost the entire text, please correct it. Same with "substantia nigra".

→ As your comments, we revised.

Line 105: please expand P. pentosaceus (Pediococcus pentosaceus)

→ We revised the word in line 110.

It should be -80 instead of _80

→ We revised the error in line 303.

I think it would be better to expand the abbreviation PD (Parkinson’s disease)

→ Thank you for your comment. We expanded the abbreviation in line 349-350.

Reviewer 4 Report

The manuscript entitled “Probiotic properties and neuroprotective effects of Lactobacillus buchneri KU200793 isolated from Korean fermented foods” submitted for revision in the International Journal of Molecular Sciences had been positively reviewed with some minor modifications.

 This is an interesting manuscript concerningprobiotic potential effects and neuroprotective effects of bacteria isolated from Korean fermented foods.

Research results expand knowledge of probiotic characteristics and neuroprotective effects of bacteria isolated from Korean fermented foods. The Authors described the potential probiotic properties such as high tolerance against artificial gastric juice and bile salts, sensitivity to antibiotics, non-production of a carcinogenic enzyme, and high adhesion to intestinal cells of three bacterial strains (Lactobacillus fermentum KU200060, Lactobacillus delbrueckii KU200171, and Lactobacillus buchneri KU200793)

The review is positive. Therefore, I propose to accept this paper for publication after minor amendments.

The title should be changed because in the manuscript was described L. fermentum KU200060, L. delbrueckii KU200171, and L. buchneri KU200793 isolated from Korean fermented foods. Maybe it should be: Probiotic properties and neuroprotective effects of L. fermentum KU200060, L. delbrueckii KU200171, and L. buchneri KU200793 isolated from Korean fermented foods

Keywords: I suggest you delete the words “probiotics”, “neuroprotective effect”. These words are already in the title of the publication and should not be repeated Line 50-54 Introduction: I suggest changed sentence and added L. fermentum KU200060, L. delbrueckii KU200171 ex.: “The purpose of this study was to identify L. buchneri KU200793…” changed on: “The purpose of this study was to identify L. fermentum KU200060, L. delbrueckii KU200171, and L. buchneri KU200793…” also should be changed: Additionally, this study investigated the neuroprotective effect of heat-killed L. buchneri KU200793 against MPP+- induced cytotoxicity in SH-SY5Y cells The order of the chapters should be changed. The order should be: Introduction, Materials and Methods, Results and Discussion, Conclusions Line 142-143: Should be described: “The viability of SH-SY5Y cells following treatment with CM containing L. rhamnosus GG, L. 142 fermentum KU200060, L. delbrueckii KU200171, and L. buchneri KU200793 was 71.96 ± 0.98%, 69.20 ± 143 0.89%, 66.81 ± 2.23%, and 73.37 ± 0.41%, respectively”. In the case of L. delbrueckii KU200171 no significant difference was noted. Line 159-161: Should be described: “The expression levels of BDNF in HT-29 cells treated with heat-killed L. rhamnosus GG, L.fermentum KU200060, L. delbrueckii KU200171, and L. buchneri KU200793 were increased by 4.55-, 1.24-, 1.36-, and 5.77-fold compared to the levels in the negative control, respectively”. In the case of L. fermentum KU200060 no significant difference was noted. Line 189-192: Should be described: “However, L. buchneri KU200793 significantly decreased the Bax/Bcl-2 ratio to almost normal levels (by 1.02-fold). L. rhamnosus GG, L. fermentum KU200060, and L. delbrueckii KU200171 decreased the ratio by 1.49-, 1.88-, and 2.23-fold, respectively”. In the case of L. buchneri KU200793 and L. rhamnosus GG no significant difference was noted and also in the case L. fermentum KU200060 L. delbrueckii KU200171 no significant difference was noted. The authors should be check that all the abbreviations in the text have been explained. I suggest adding a list of abbreviations used at the beginning of the publication

Author Response

- This is an interesting manuscript concerning probiotic potential effects and neuroprotective effects of bacteria isolated from Korean fermented foods.

Research results expand knowledge of probiotic characteristics and neuroprotective effects of bacteria isolated from Korean fermented foods. The Authors described the potential probiotic properties such as high tolerance against artificial gastric juice and bile salts, sensitivity to antibiotics, non-production of a carcinogenic enzyme, and high adhesion to intestinal cells of three bacterial strains (Lactobacillus fermentum KU200060, Lactobacillus delbrueckii KU200171, and Lactobacillus buchneri KU200793).

The review is positive. Therefore, I propose to accept this paper for publication after minor amendments.

→ Thank you for your considerations and comments.

1.The title should be changed because in the manuscript was described L. fermentum KU200060, L. delbrueckii KU200171, and L. buchneri KU200793 isolated from Korean fermented foods. Maybe it should be: Probiotic properties and neuroprotective effects of L. fermentum KU200060, L. delbrueckii KU200171, and L. buchneri KU200793 isolated from Korean fermented foods.

→ Thanks, we represented the candidates of three types of lactic acid bacteria (L.fermentum 20060, L. delbrueckii 200171, L. buchneri 200793). However, all of their heat-killed forms have not neuroprotective effects. Therefore, the manuscript title seems appropriate to mention only L. buchneri KU200793.

2 Keywords: I suggest you delete the words “probiotics”, “neuroprotective effect”. These words are already in the title of the publication and should not be repeated Line 50-54

→ Keywords can suggest represented words. So, we did not changed.

Introduction: I suggest changed sentence and added L. fermentum KU200060, L. delbrueckii KU200171. ex) “The purpose of this study was to identify L. buchneri KU200793…”

changed on: “The purpose of this study was to identify L. fermentum KU200060, L. delbrueckii

KU200171, and L. buchneri KU200793…” also should be changed: Additionally, this study investigated the neuroprotective effect of heat-killed L. buchneri KU200793 against MPP+- induced cytotoxicity in SH-SY5Y cells

→ Thank you for your careful comment, we revised in line 50.

The order of the chapters should be changed. The order should be: Introduction, Materials and Methods, Results and Discussion, Conclusions

→ This manuscript writing referred order is : Introduction, Results and Discussion, Materials and Methods and Conclusions by International Journal of Molecular Sciens.

Line 142-143: Should be described: “The viability of SH-SY5Y cells following treatment with CM containing L. rhamnosus GG, L. fermentum KU200060, L. delbrueckii KU200171, and L. buchneri KU200793 was 71.96 ± 0.98%, 69.20 ± 0.89%, 66.81 ± 2.23%, and 73.37 ± 0.41%, respectively”. In the case of L. delbrueckii KU200171 no significant difference was noted.

→ As you suggested, The S.D. values of L. delbrueckii KU200171 was higher than other data. However, this data is significant in ANOVA analysis. Therefore, we did not represent.

민정아, 이 값이 no significant 하니?

Line 159-161: Should be described: “The expression levels of BDNF in HT-29 cells treated with heat-killed L. rhamnosus GG, L.fermentum KU200060, L. delbrueckii KU200171, and L. buchneri KU200793 were increased by 4.55-, 1.24-, 1.36-, and 5.77-fold compared to the levels in the negative control, respectively”. In the case of L. fermentum KU200060 no significant difference was noted.

→ Similar to cell viability experiments, In BDNF RNA expression experiments was no statistically significant difference except for L. rhamnosus GG and L. buchneri KU200793 strain, Therefore the manuscript mentioned that only L. buchneri KU200793 was effective.

Line 189-192: Should be described: “However, L. buchneri KU200793 significantly decreased the Bax/Bcl-2 ratio to almost normal levels (by 1.02-fold). L. rhamnosus GG, L. fermentum KU200060, and L. delbrueckii KU200171 decreased the ratio by 1.49-, 1.88-, and 2.23-fold, respectively”. In the case of L. buchneri KU200793 and L. rhamnosus GG no significant difference was noted and also in the case L. fermentum KU200060 L. delbrueckii KU200171 no significant difference was noted.

→ Thank you for your comments. The LGG strain used in the experiments was showed to have a neuroprotective effect by previous studies (Cheng et al., 2019). Therefore, LGG and L. buchneri KU200793 did not show significantly difference but they are similar to Bax/Bcl-2 ratio. These results were meaningful. And L. fermentum KU200060 and L. delbruckii KU200171 were concluded that both strains were ineffective because they were not statistically different from the MPP + only group.  

7.The authors should be check that all the abbreviations in the text have been explained. I suggest adding a list of abbreviations used at the beginning of the publication

→ Thank you for your comments, we revised the list of abbreviations in line 357.

Thank you again for your helpful comments.

Round 2

Reviewer 2 Report

The authors improved their manuscript according to reviewers' comments.

In addition all comments were replied.